# SAQNN: Spectral Adaptive Quantum Neural Network as a Universal Approximator

**Jialiang Tang** [1 2]    **Jialin Zhang** [1 2]    **Xiaoming Sun** [1 2]

## Abstract

Quantum machine learning (QML), as an interdisciplinary field bridging quantum computing and machine learning, has garnered significant attention in recent years. Currently, the field as a whole faces challenges due to incomplete theoretical foundations for the expressivity of quantum neural networks (QNNs). In this paper we propose a constructive QNN model and demonstrate that it possesses the universal approximation property (UAP), which means it can approximate any square-integrable function up to arbitrary accuracy. Furthermore, it supports switching function bases, thus adaptable to various scenarios in numerical approximation and machine learning. Our model has asymptotic advantages over the best classical feed-forward neural networks in terms of circuit size and achieves optimal parameter complexity when approximating Sobolev functions under $L_2$ norm.

## 1. Introduction

Quantum computing is a novel computational paradigm leveraging the high-dimensionality of Hilbert spaces to perform computations that are intractable for classical devices. While algorithms such as Shor's algorithm for integer factorization (1997) and Grover's algorithm for unstructured search (1996) have demonstrated quantum advantages, a broader question remains: can this computational power be harnessed for machine learning? This inquiry has given rise to Quantum Machine Learning (QML), a field dedicated to investigating the performance of quantum models in machine learning tasks (Schuld et al., 2015; Biamonte

et al., 2017; Dallaire-Demers & Killoran, 2018; Schuld et al., 2020; Cerezo et al., 2021; Huang et al., 2021; Ye et al., 2025).

Over the past decades, machine learning has achieved remarkable success, fundamentally reshaping modern technology. Models constructed on classical architectures, including deep Feed-Forward Networks (FNNs) (Rumelhart et al., 1986; Bebis & Georgiopoulos, 1994; Eldan & Shamir, 2016; Qamar & Zardari, 2023; Aftabi et al., 2025) to recent advancements in diffusion models (Sohl-Dickstein et al., 2015; Ho et al., 2020; Song et al., 2021; He et al., 2025) and Large Language Models (LLMs) (Vaswani et al., 2017; Brown et al., 2020; Ouyang et al., 2022; Naveed et al., 2025), are now capable of executing complex tasks like classification, clustering, and high-fidelity generation. The success is underpinned by a mature theoretical framework, most notably the Universal Approximation Theorem (UAT) (Cybenko, 1989; Hornik et al., 1989). This theorem guarantees that classical neural networks, given sufficient width or depth, can approximate continuous functions to arbitrary accuracy, providing a solid mathematical justification for their expressive power.

Quantum Neural Networks (QNNs) which are typically realized as parameterized quantum circuits (PQCs) have been deployed across a vast array of learning tasks in recent years (Schuld et al., 2014; Killoran et al., 2019; Jia et al., 2019; Abbas et al., 2021; Huang et al., 2023), from image classification (Farhi & Neven, 2018; Schuld et al., 2019; Mathur et al., 2021; Shi et al., 2024) to solving physical systems (Li & Benjamin, 2017; Kandala et al., 2017; Yuan et al., 2019; Lubasch et al., 2020; Kyriienko et al., 2021). Despite this empirical enthusiasm, the theoretical foundations of QNNs remain underdeveloped compared to the mature approximation theory of classical neural networks. A significant portion of QNNs research relies on heuristic network designs, where the relationship between the circuit architecture and the function space it can approximate is poorly understood. This leads to a critical bottleneck: without a rigorous understanding of the relations between model structure and its expressivity, it is difficult to determine whether a quantum model offers genuine advantages over classical models like deep neural networks. Also, how to construct

[1]State Key Lab of Processors, Institute of Computing Technology, Chinese Academy of Sciences, Beijing 100190, China [2]School of Computer Science and Technology, University of Chinese Academy of Sciences, Beijing 100049, China. Correspondence to: Jialiang Tang <tangjialiang20@mails.ucas.ac.cn>, Xiaoming Sun <sunxiaoming@ict.ac.cn>.

*Proceedings of the 43rd International Conference on Machine Learning*, Seoul, South Korea. PMLR 306, 2026. Copyright 2026 by the author(s).

powerful QNNs theoretically reliably is a question.

The central challenge is to establish a constructive approximation theory for QNNs. Unlike classical neural networks, QNNs must use quantum processes like measurement or data re-uploading schemes to realize the approximation while adhering to unitary constraints. Furthermore, for a QNN to be practically viable, it is insufficient to merely prove that it can approximate functions, the more important thing is to quantify the cost of circuit. This paper addresses these challenges by proposing a constructive QNN architecture inspired by linear combination of unitary (LCU), providing both universality guarantees and error bounds for approximating functions in Sobolev spaces under $L_2$ norm.

## 1.1. Related Research

**Classical Models**. In the classical regime, the expressive power of neural networks is well-established. The seminal works by Cybenko (Cybenko, 1989) and Hornik (Hornik et al., 1989) proved that feed-forward networks with a single hidden layer possess Universal Approximation Property (UAP) for continuous functions. Beyond mere existence, recent research has focused on quantitative analysis, establishing rigorous error bounds and convergence rates for various function spaces. Notably, the approximation capabilities of FNNs have been characterized for continuous functions (Shen et al., 2019; 2020; 2022; Yarotsky, 2018), Sobolev functions (Yarotsky, 2017; Yarotsky & Zhevnerchuk, 2020; Lu et al., 2021; Yang, 2025) and Besov functions (Siegel, 2023; Suzuki, 2019; Yang, 2025) in terms of circuit width, circuit depth and parameter complexity.

**Quantum-Classical Hybrid Models**. In the quantum domain, several recent studies have begun to address these foundational questions. The UAP for quantum-classical hybrid networks is demonstrated (Goto et al., 2021; Hou et al., 2023), while another work shows even one qubit is sufficient to approximate any bounded function (Pérez-Salinas et al., 2021). However, these architectures face limitations: the expressivity of hybrid model comes mostly from classical processes and the QNN only serves as the activation function.

**Quantum Models**. Efforts have been made to investigate power of quantum models without any classical trainable process. Schuld et al. (2021) proposed a multi-qubit model, implementing truncated Fourier series to obtain UAP. The qubit cost was optimized in later work (Pérez-Salinas et al., 2025). But it relies on arbitrary global gates and parameterized observable, the latter of which requires complicated classical post-process. Thus the model is non-constructive, impractical and essentially hybrid. For univariate functions, a notable progress is that single-qubit model is sufficient for UAP (Yu et al., 2022), but it does not fully utilize the entanglement power of quantum computing and encounters

difficulties in approximating multivariate functions. The first constructive quantum model that has UAP for multivariate functions was proposed by Yu et al. (2024). They also provided non-asymptotic error bounds for Lipschitz functions and Hölder functions under $L_\infty$ norm. However, the $L_\infty$ accuracy is often too stringent in practical learning tasks dominated by average-case performance, and the power of QNN approximating more function spaces remains unknown. This is the theoretical gap our work aims to fill.

## 1.2. Our Results

In this paper, we propose a constructive QNN model named **Spectral Adaptive Quantum Neural Network (SAQNN)**, of which the architecture is shown in Figure 1. We rigorously prove its UAP for arbitrary square-integrable functions and establish error bounds for $L_2$-approximation of Sobolev functions. Our analysis characterizes the asymptotic resource costs in terms of circuit width (number of qubits), circuit depth, and parameter complexity—metrics analogous to the network width and depth of classical neural networks. Finally, numerical experiments are conducted to verify the feasibility of our model. We now state the main theorems.

Denote the quantum circuit of SAQNN as $U_{\boldsymbol{\theta},\boldsymbol{\phi}}(\boldsymbol{x})$, in which $\boldsymbol{\theta}$, $\boldsymbol{\phi}$ are vectors of trainable parameters in state preparation and phase injection, and $\boldsymbol{x} = (x_1, \ldots, x_d)$ is $d$-dimensional inputs encoded in rotation gates.

**Theorem 1.** *For any multivariate square-integrable function $f : [-\pi, \pi]^d \to [0, 1]$ and any accuracy $\epsilon$, there exists a SAQNN $U_{\boldsymbol{\theta},\boldsymbol{\phi}}(\boldsymbol{x})$ with parameters $\boldsymbol{\theta}$, $\boldsymbol{\phi}$ and rescale coefficient $a$ s.t.*

$$\|a \langle 0| U_{\boldsymbol{\theta},\boldsymbol{\phi}}(\boldsymbol{x})^{\dagger} O U_{\boldsymbol{\theta},\boldsymbol{\phi}}(\boldsymbol{x}) |0\rangle^{1/2} - f(\boldsymbol{x})\|_2 \leq \epsilon,$$

*where the global observable $O = |0\rangle \langle 0|$.*

In order to quantify the approximation cost of our QNN, we target Sobolev function spaces, a fundamental function class the field of classical neural network approximation concerns a lot (Yarotsky, 2017; Lu et al., 2021). Sobolev spaces are inclusive, naturally embedding broad function classes such as continuous functions (ada, 2003). Furthermore, since solutions to many physical systems inherently reside in Sobolev spaces (Evans, 2022), establishing theoretical guarantees here suggests strong potential for quantum learning in physical tasks. Crucially, the Fourier series, which constitutes the optimal approximation basis for Sobolev functions (Pinkus, 2012), can be naturally implemented on quantum circuits.

**Theorem 2.** *For any multivariate Sobolev function $f \in H_u^s([-\pi, \pi]^d)$ with value range $[0, 1]$ and any $\epsilon > 0$, there exists a SAQNN $U_{\boldsymbol{\theta},\boldsymbol{\phi}}(\boldsymbol{x})$ with parameters $\boldsymbol{\theta}$, $\boldsymbol{\phi}$ and rescale coefficient $a$ s.t.*

$$\|a \langle 0| U_{\boldsymbol{\theta},\boldsymbol{\phi}}(\boldsymbol{x})^{\dagger} O U_{\boldsymbol{\theta},\boldsymbol{\phi}}(\boldsymbol{x}) |0\rangle^{1/2} - f(\boldsymbol{x})\|_2 \leq \epsilon,$$

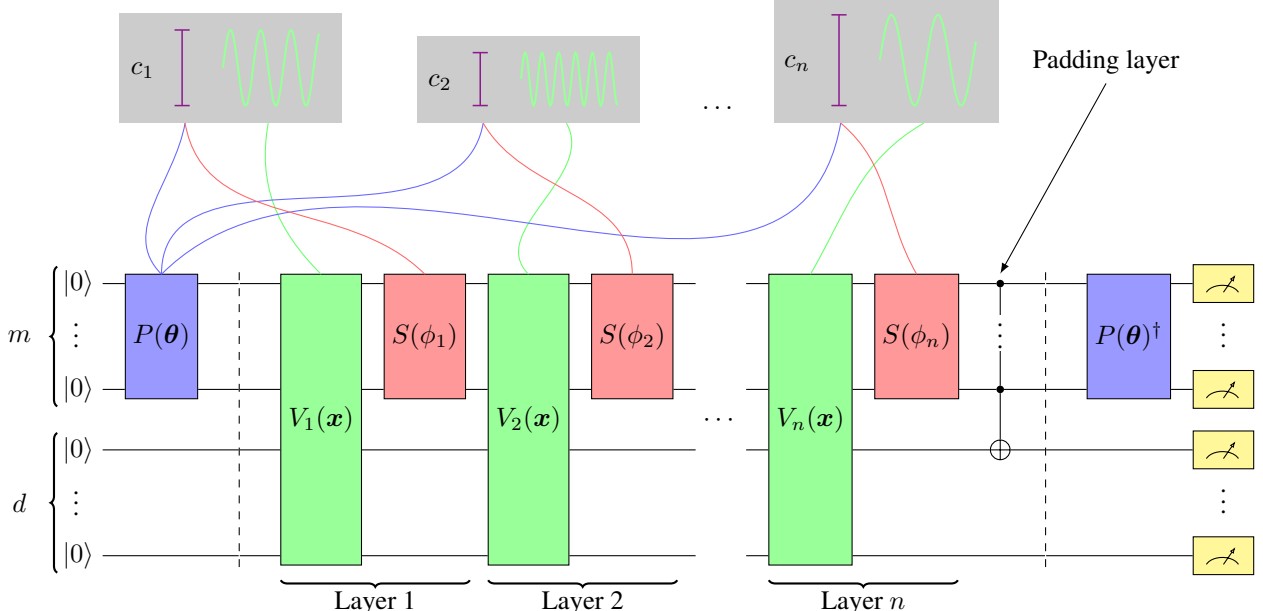

*Figure 1.* The construction of SAQNN. The state preparation block, spectrum selection block and phase injection are marked blue, green and red respectively. When approximating functions of $d$ variables using $n$ terms of truncated Fourier series, the whole circuit consists of $m + d$ qubits, with state preparation block $P(\boldsymbol{\theta})$, $n$ layers of spectrum selection block accompanied by phase injection, a padding layer and an inversion of $P(\boldsymbol{\theta})$ applied before final measurement. Here $m = \lceil \log n \rceil$ and $c_1, c_2, \cdots, c_n$ denotes coefficients of Fourier series with $n$ terms.

*where the global observable $O = |0\rangle \langle 0|$. The circuit width (number of qubits) of $U_{\boldsymbol{\theta}, \boldsymbol{\phi}}(\boldsymbol{x})$ is $O(\log n)$, the circuit depth is $O(n \log n)$ and the parameter complexity is $O(n)$. Here $n = O((d + 1/\epsilon)^{16(1/\epsilon)^{2/s}})$.*

A more compressive encoding strategy is adopted in analysis of Theorem 2, optimizing number of qubits from $O(\log n + d)$ to $O(\log n)$, see Appendix D. Note that we are the first to analyze the error bounds of QNNs approximating Sobolev functions under $L_2$ norm in terms of circuit width, circuit depth and parameter complexity.

Besides UAP, our model has more advantages:

- It is a constructive model, which means the construction (or synthesis) of circuit is explicit with basic gates, distinguished from the model with non-constructive global blocks proposed by Schuld (2021).

- It supports basis shift between Fourier series and Chebyshev series, thus adaptive to various scenarios in numerical approximation and machine learning.

- On $L_2$-approximating of Sobolev functions, it has quantum advantages over State-Of-The-Art (SOTA) of classical neural networks in $d$-demanding case (with constant accuracy $\epsilon$ and smoothness $s$).

- On $L_2$-approximation of Sobolev functions, it achieves the optimal parameter complexity with regards to the

asymptotic order of accuracy $\epsilon$ (with constant input dimension $d$ and smoothness $s$).

To clarify our contribution, we not only propose a constructive QNN architecture possessing the universal approximation property but also establish rigorous error bounds for approximating Sobolev functions under $L_2$ norm. Given the current underdeveloped state of quantum machine learning theory, this work advances the theoretical understanding of QNNs' expressivity. Since Fourier and Chebyshev bases are fundamental in approximation theory. Our constructive QNN model provides explicit implementation of these bases on quantum circuits, offering a versatile and interpretable framework for machine learning and function approximation. By enabling efficient approximation with these canonical bases, our model offers high-level guidance for design of QNNs in practical scenarios, paving the way for more reliable and powerful quantum models.

### 1.3. Organization

The rest of the paper is organized as below. Section 2 is about the preliminaries. In section 3, we introduce our model construction in detail, then provide analysis of error bounds approximating Sobolev functions. The main results are placed here. In section 4 we conduct numerical experiments to verify the feasibility of our model, while proposing a practical scaling strategy. We make a summary of this

paper in section 5, including achievements and drawbacks, and then raise some open problems for future work.

## 2. Preliminaries

Given a matrix $U$, we denote its $(i,j)$-th entry as $U_{ij}$, and the conjugate transpose of $U$ as $U^\dagger$. We use $\otimes$ to denote the Kronecker tensor product over two matrices.

### 2.1. Basic Knowledge of Quantum Computing

**Quantum State.** The basic computing unit in quantum computation is qubit. It is a unit vector in Hilbert space $\mathcal{H} \cong \mathbb{C}^2$ and its state can be denoted by $|\psi\rangle = \alpha|0\rangle + \beta|1\rangle$, which satisfies the normalization condition $|\alpha|^2 + |\beta|^2 = 1$. A quantum state of $n$ qubits is a vector in $n$-product Hilbert spaces $\mathcal{H}^n \cong \mathbb{C}^{2^n}$. Unlike that in classical computing, the state can be in exponential basic states simultaneously and the coefficients can be understood as probability amplitude. We denote $\langle\psi|$ as the conjugate transpose of $|\psi\rangle$ and inner product of two states $|\varphi\rangle$ and $|\psi\rangle$ is $\langle\varphi|\psi\rangle$. These are well-known *Dirac notation*.

**Quantum Gate.** Quantum gate is the computation (or evolution) operated on quantum qubits. It's a unitary matrix $U \in U(2^n)$. The most used gates include 2-qubit gate $CNOT = I_2 \oplus X$, single-qubit rotation gates $R_x(\theta) = e^{-\theta X/2}$, $R_y(\theta) = e^{-\theta Y/2}$, $R_z(\theta) = e^{-\theta Z/2}$. The $X, Y, Z$ are Pauli operators defined as

$$X \equiv \begin{pmatrix} 0 & 1 \\ 1 & 0 \end{pmatrix}, Y \equiv \begin{pmatrix} 0 & -i \\ i & 0 \end{pmatrix}, Z \equiv \begin{pmatrix} 1 & 0 \\ 0 & -1 \end{pmatrix}.$$

**Measurement.** A measurement is a quantum process extracting classical information from qubits. The result of measurement on $|\psi\rangle$ with observable $O$ is $\langle\psi|O|\psi\rangle$, where $O$ satisfies Hermitian condition $O = O^\dagger$. Since quantum measurement is probabilistic, the result is an expectation value.

### 2.2. Quantum Multiplexor

A quantum multiplexor (Shende et al., 2006) with $s$ controlled qubits (positioned at the highest levels) and $d$ target qubits is a block-diagonal unitary matrix comprising $2^s$ unitary matrices $U_i \in U(2^d)$, $1 \le i \le 2^s$:

$$U = \begin{pmatrix} U_1 & & \\ & \ddots & \\ & & U_{2^s} \end{pmatrix}.$$

Here is a typical example: the multiplexor-$R_y$, which is a multiplexor with 1 data qubit acted by $R_y$ gate with different angles depending on the state of controlled qubits. In the circuit diagram, we mark each controlled qubit of $U$ in quantum circuits with a "□" symbol.

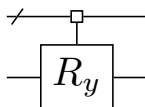

**Lemma 3** (Decomposition of multiplexor rotation gates (Bullock & Markov, 2003; Möttönen et al., 2004; Shende et al., 2006)). *For multiplexor-$R_k$ and $k = y, z$, it holds that*

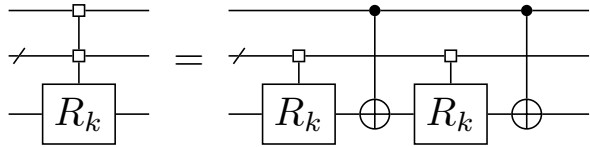

*and any $n$-qubit multiplexor-$R_k$ can be synthesized by at most $2^{n-1}$ CNOT gates.*

One should distinguish between multiplexor gates with controlled-$U$ gates. The latter is to apply $U$ under a specific state of control qubits, but do nothing otherwise.

### 2.3. Data Re-uploading Quantum Neural Networks

In quantum neural networks, how the inputs come and be encoded will fundamentally affect it's expressivity. The commonly used encoding strategy is *data re-uploading* (Pérez-Salinas et al., 2020; Schuld et al., 2021; Wach et al., 2023), which encodes the inputs into angles of rotation gates. An example of 2-qubit QNN model is shown below.

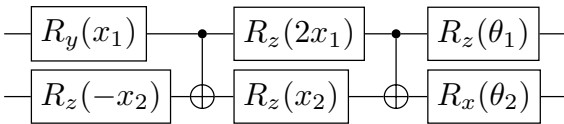

In the circuit we can upload the inputs $x_1, x_2$ repeatedly, and train the parameters $\theta_1, \theta_2$. Note that simple classical pre-process is allowed since it's not parameterized, distinguished from hybrid networks.

## 3. Model Construction and Analysis

A useful mathematical tool in approximation theory is *Fourier series*. In many natural settings, a Fourier series of a univariate function $f(x)$ is the linear combination of periodic function bases with integer frequencies:

$$f(x) = \sum_{j=-\infty}^{\infty} c_j e^{ijx}.$$

More generally, the Fourier series for multivariate functions $f(\boldsymbol{x})$ works in a similar way,

$$f(\boldsymbol{x}) = \sum_{j_1,\dots,j_d=-\infty}^{\infty} c_j e^{i\boldsymbol{j}\boldsymbol{x}},$$

where the frequency $\boldsymbol{j} = (j_1, j_2, \cdots, j_d)$ and the inputs $\boldsymbol{x} = (x_1, x_2, \cdots, x_d)$ are $d$-dimension vectors.

A *truncated Fourier series* keep finite items in Fourier series, usually the function bases with degree $\|\boldsymbol{j}\|_\infty \le k$, and noted as

$$f_k(\boldsymbol{x}) = \sum_{j_1,\ldots,j_d=-k}^{k} c_{\boldsymbol{j}} e^{i\boldsymbol{j}\boldsymbol{x}}.$$

It's an important result in approximation theory that any square-integrable functions can be approximated by a truncated Fourier series to arbitrary accuracy.

**Lemma 4** (Fourier approximation (Stein & Weiss, 1971; Rivlin, 1981; Weisz, 2012))**.** *For any* $f \in L_2([-\pi, \pi]^d)$ *and* $\epsilon > 0$*, there exists a* $k$*-truncated Fourier series* $f_k$ *s.t.*

$$\|f(\boldsymbol{x}) - f_k(\boldsymbol{x})\|_2 \le \epsilon.$$

We now move to the details of our model. We investigate how our model implements a truncated Fourier series, and how the different components of SAQNN circuit control the expressivity.

### 3.1. Structure of SAQNN

The Spectral Adaptive Quantum Neural Network is composed of three main parts: state preparation, spectrum selection and phase injection. These components are marked in colors in Figure 1. The design of our model is inspired from linear combination of unitaries (LCU) (Childs & Wiebe, 2012; Gilyén et al., 2019), which is widely used in Hamiltonian simulation (Berry et al., 2015; Chakraborty, 2024).

We first provide intuition behind the model construction and its approximation properties. A Fourier series is expressed as $f(\boldsymbol{x}) = \sum_{\boldsymbol{j}} |c_{\boldsymbol{j}}| e^{i \arg(c_{\boldsymbol{j}})} e^{i\boldsymbol{j}\cdot\boldsymbol{x}}$. The SAQNN architecture essentially deconstructs this mathematical formulation and maps its components directly onto a quantum circuit. Specifically, the state preparation $P(\boldsymbol{\theta})$ corresponds to the amplitude $|c_{\boldsymbol{j}}|$, mapping the coefficient magnitudes one-to-one to the probability amplitudes of the control qubits. Subsequently, the phase injection $S(\boldsymbol{\phi})$ acts as the phase term $\arg(c_{\boldsymbol{j}})$, attaching global phases to each magnitude and thereby equipping the model with the capacity to learn complex coefficients. Finally, the spectrum selection $V_r(\boldsymbol{x})$ represents the basis functions $e^{i\boldsymbol{j}\cdot\boldsymbol{x}}$, encoding the inputs via quantum rotation gates. Since a standard Linear Combination of Unitaries (LCU) framework block-encodes the linear combination of conditional operations,

$$U_{\boldsymbol{\theta},\boldsymbol{\phi}}(\boldsymbol{x}) = \begin{pmatrix} \sum_r c_r U_r(\boldsymbol{x}) & * \\ * & * \end{pmatrix},$$

it inherently leads to the approximation properties.

### 3.1.1. STATE PREPARATION

In typical LCU settings, the state preparation step is to prepare arbitrary states on $|0\rangle$. But here we adopt the circuit with multiplexor-$R_y$ gates to implement states with arbitrary real amplitudes.

The circuit of state preparation is shown below. Note that for the sake of simplicity, we have omitted the specific forms of the parameters, but we always provide a detailed explanation of how the parameters are encoded into the quantum gates.

It is composed of multiplexor-$R_y$ gates on the first $i$ qubits for $1 \le i \le m$. The $i$-th gate contains $2^{i-1}$ parameters, serving as the rotation angles in $2^{i-1}$ cases of control qubits. Denote the $i$-th multiplexor-$R_y$ as $MR_y^{(i)}$, then

$$MR_y^{(i)} = \begin{pmatrix} R_y(\theta_{2^{i-1}}) & & \\ & \ddots & \\ & & R_y(\theta_{2^i-1}) \end{pmatrix}.$$

This block is key to the parameter optimality of our model, it generates arbitrary states with real amplitudes and with phase injection contributes to generation of coefficients in Fourier series. Appendix A and D shows how that works.

### 3.1.2. SPECTRUM SELECTION

In spectrum selection we encode inputs into single-qubit rotation gates to implement the items of truncated Fourier series. To be specific, each $V_r(\boldsymbol{x})$ gate is in the form of a controlled-$U_r$ gates.

$V_r(\boldsymbol{x})$ applies a $U_r(\boldsymbol{x})$ to the bottom $d$ qubits under the condition of top $m$ qubits being $|r\rangle$.

For construction of $U_r$, we use the tensor product of $R_z$ gates, encoding the inputs and implement each basis of multivariate Fourier series.

In matrix representation,

$$U_r(\boldsymbol{x}) = R_z(-2j_1^{(r)}x_1)\otimes R_z(-2j_2^{(r)}x_2)\otimes R_z(-2j_d^{(r)}x_d),$$

with the equation holds that

$$\langle 0|^{\otimes d}\, U_r\, |0\rangle^{\otimes d} = e^{i\boldsymbol{j}^{(r)}\boldsymbol{x}},$$

$\boldsymbol{j}^{(r)} = (j_1^{(r)}, j_2^{(r)}, \cdots, j_d^{(r)})$ is the $r$-th frequency in Fourier series, satisfying $1 \le r \le n$.

Also, the controlled-$U_r$ can be equally represented by a sequence of "stair" controlled-$R_z$ gates.

The controlled-$R_z$ gates can be synthesized by two multi-qubit Toffoli gates and two $R_z$ gates, contributing to the analysis of circuit depth in Theorem 2.

3.1.3. PHASE INJECTION

To equip our model with the expressivity for complex numbers, we introduce phase injection technique into spectrum selection part. For each controlled layer inside, we apply a controlled phase gate to provide global phase for $U_r$, which is parameterized for training. That is,

The representation is

$$S(\phi_r) = e^{i\phi_r}|r\rangle\langle r| + \sum_{j\ne r}|j\rangle\langle j|.$$

As shown in Figure 1, our model starts on $|0\rangle$, chronologically applying a state preparation block, $n$ layers of spectrum selection block with phase injection, a multi-qubit Toffoli gate as padding layer and the inversion of state preparation block. The result is measured with a global observable

$O = |0\rangle\langle 0|$. Here the state preparation block and phase injection are parameterized to for training and the inputs are encoded into spectrum selection block. We will see such a model can work as a universal approximator.

### 3.2. Universal Approximation Properties

To meticulously describe the UAP of our model, we restate Theorem 1, which indicates SAQNN can approximate any multivariant square-integrable functions under arbitrary accuracy. To clarify, we choose the $L_2$ norm since it is directly related to the Mean Squared Error (MSE), which is a widely-used loss function in practical machine learning tasks. As an average-case metric, the $L_2$ norm may under-penalize localized predictive errors (e.g., local spikes), which could affect point-wise reliability.

**Theorem 1.** *For any multivariate square-integrable function $f : [-\pi, \pi]^d \to [0, 1]$ and any accuracy $\epsilon$, there exists a SAQNN $U_{\boldsymbol{\theta},\boldsymbol{\phi}}(\boldsymbol{x})$ with parameters $\boldsymbol{\theta}$, $\boldsymbol{\phi}$ and rescale coefficient $a$ s.t.*

$$\|a\,\langle 0|\,U_{\boldsymbol{\theta},\boldsymbol{\phi}}(\boldsymbol{x})^\dagger O U_{\boldsymbol{\theta},\boldsymbol{\phi}}(\boldsymbol{x})\,|0\rangle^{1/2} - f(\boldsymbol{x})\|_2 \le \epsilon,$$

*where the global observable $O = |0\rangle\langle 0|$.*

The proof of Theorem 1 is placed in Appendix A. Remark that $O(1/\epsilon^2)$ times of measurement is needed to estimate the expectation value up to an additive accuracy $\epsilon$. But we can further use the amplitude estimation (Brassard et al., 2002) to reduce the repetitive times to $O(1/\epsilon)$ with increasing circuit depth by $O(1/\epsilon)$. The classical square-root post-process on the expectation output is taken in our theorems, which extracts the amplitudes of final $|0\rangle$ state. The operation is physically reasonable since it does essentially the same thing as amplitude estimation.

The rescale coefficient $a$ in Theorem 1 is not a parameter to be classically trained, distinguishing SAQNN from hybrid model. It can be bounded and a fine-tuning strategy for numerical feasibility is introduced in Appendix B. In Appendix B, one can even see it's theoretical reliable to fix $a$ to a specific value all the way, which doesn't affect the correctness of Theorem 1.

Although Fourier series have outstanding performance in approximation of periodic functions, it's easy to encounter problems in non-periodic cases (which is called *Gibbs phenomenon* (Gottlieb & Shu, 1997) in approximation theory). However, our model has good extensibility. The function basis implemented can be shifted to Chebyshev series naturally in SAQNN to adapt to various scenarios in numerical approximation and machine learning. Details are presented in Appendix C.

In our model, each layer of spectrum selection block controls a Fourier basis with specific frequency. The state preparation block and phase injection generate complex

coefficients in Fourier series together, with the former determining modulus and the latter adjusting phases. The relations are also pictured in Figure 1. It can be seen that our model has adjustable capability by changing the layers of spectrum selection, with more rich accessible frequency spectrum contributing to more powerful expressivity.

## 3.3. Error Bounds for Sobolev Functions

Theorem 1 establishes the expressivity of SAQNN, but the circuit cost needed is absent. We go further to give an explicit analysis and show how SAQNN has advantages over SOTA of classical neural networks under reasonable settings. In this paper we consider the Sobolev spaces, results in more function spaces remain open problems.

**Definition 5** (Sobolev function spaces (ada, 2003; Canuto et al., 2006)). For $s \in \mathbb{N}$, Sobolev function space $H^s([-\pi, \pi]^d)$ is defined as

$$H^s([-\pi, \pi]^d) := \{f : D^{\boldsymbol{\alpha}} f \in L^2([-\pi, \pi]^d), \forall \|\boldsymbol{\alpha}\|_1 \le s\},$$

where

$$D^{\boldsymbol{\alpha}} f = \frac{\partial f}{\partial^{\alpha_1} x_1 \partial^{\alpha_2} x_2 \cdots \partial^{\alpha_d} x_d}.$$

$H^s([-\pi, \pi]^d)$ is a normed space equipped with Sobolev norm

$$\|f\|_{H^s} = \left(\sum_{|\boldsymbol{\alpha}| \le s} \|D^{\boldsymbol{\alpha}} f\|_2^2\right)^{1/2}.$$

In usual settings of approximation theory, the Sobolev unit ball $f \in H_u^s([-\pi, \pi]^d)$ where $\|f\|_{H^s} \le 1$ is considered in analysis of error bounds. Similar to that in classical neural networks, we focus on the resource cost to be circuit width (number of qubits), circuit depth and number of parameters.

**Theorem 2.** *For any multivariate Sobolev function $f \in H_u^s([-\pi, \pi]^d)$ with value range $[0, 1]$ and any $\epsilon > 0$, there exists a SAQNN $U_{\boldsymbol{\theta}, \boldsymbol{\phi}}(\boldsymbol{x})$ with parameters $\boldsymbol{\theta}$, $\boldsymbol{\phi}$ and rescale coefficient $a$ s.t.*

$$\|a \langle 0| U_{\boldsymbol{\theta}, \boldsymbol{\phi}}(\boldsymbol{x})^{\dagger} O U_{\boldsymbol{\theta}, \boldsymbol{\phi}}(\boldsymbol{x}) |0\rangle^{1/2} - f(\boldsymbol{x})\|_2 \le \epsilon,$$

*where the global observable $O = |0\rangle \langle 0|$. The circuit width (number of qubits) of $U_{\boldsymbol{\theta}, \boldsymbol{\phi}}(\boldsymbol{x})$ is $O(\log n)$, the circuit depth is $O(n \log n)$ and the parameter complexity is $O(n)$. Here $n = O((d + 1/\epsilon)^{16(1/\epsilon)^{2/s}})$.*

The proof of Theorem 2 is placed in Appendix D.

To show our advantages, we make a comparison between the SAQNN and the SOTA of classical neural networks. For classical model, we focus on the feed-forward neural networks (FNNs) (Rumelhart et al., 1986; Bebis & Georgiopoulos, 1994; Svozil et al., 1997) with rectified linear unit (ReLU) activation function (Nair & Hinton, 2010; Glorot et al., 2011), where ReLU$(x) := \max\{0, x\}$.

Such neural networks can fundamentally represent a vast range of classical neural networks. The function spaces $H_u^s([-\pi, \pi]^d)$ are selected to be the benchmark, and the comparison is made from circuit size (defined as product of circuit width and circuit depth) and parameter complexity perspectives. To our best knowledge, the best circuit size and parameter complexity of $L_2$-approximation of Sobolev spaces for FNNs are $O((2\pi)^{3d^2/4s}(1/\epsilon)^{d/2s})$ and $O((2\pi)^{3d^2/2s}(1/\epsilon)^{d/s})$, adapted from (Yang, 2025). One can also refer to (Lu et al., 2021) and (Jiao et al., 2021) for further understanding. In empirical sense, it's often satisfied in real-world learning tasks that $s$ is a small constant but $d$ can be large (e.g. $d = 3072$ for CIFAR-10 (Krizhevsky et al., 2009), $d = 150528$ for ImageNet (Deng et al., 2009) and $d = 2408448$ for Kinetics-400 dataset (Carreira & Zisserman, 2017)). In the context of these machine learning tasks, the primary computational bottleneck arises from the input dimension $d$ rather than the approximation accuracy $\epsilon$. So we treat the target approximation error $\epsilon$ as a fixed constant (for instance, $\epsilon = 0.01$). One can see FNNs suffer from curse of dimensionality of $d$ but the costs of our model are all in poly$(d)$ by Theorem 2. That yields the quantum advantages of SAQNN over classical neural networks when approximating high-dimensional functions under certain accuracy $\epsilon$ not so small.

To make our comparison comprehensive, we take the opposite scenario into consideration that pursuing accuracy under a fixed $d$. We show in this case our model has optimal parameter complexity among all linear and non-linear approximation methods. Details for the comparison are explained in Appendix E and we give an overview here. Fixing $d$, the lower bounds of parameters to $L_2$-approximating Sobolev function spaces with linear combination of function bases (known as linear methods (Pinkus, 2012)) and continuous manifold (known as non-linear methods (DeVore, 1998), a typical example of which is classical feed-forward neural networks) are the same with regards to $\epsilon$. Concerning the optimal approximation for Sobolev spaces is Fourier series with lowest frequencies (Mallat, 1999; Pinkus, 2012), which can be implemented by SAQNN explicitly, our model achieves optimal parameter complexity among all linear and non-linear approximation methods.

## 4. Numerical Experiments

In this section, we conduct numerical experiments to verify the feasibility of our model. We focus on the case of $d = 2$, the target functions are

$$f_1(x_0, x_1) = e^{-(\sin^2(x_0/2) + \sin^2(x_1/2))},$$

$$f_2(x_0, x_1) = \frac{\cos(x_0 + x_1 + \pi/6) + 1}{2.5}$$

for Fourier series, and

$$f_3(x_0, x_1) = \frac{(x_0 - x_1 + 1)^2}{9}$$

for Chebyshev series. We sample 200 points $(x_0, x_1)$ from intervals $[-\pi, \pi]^2$ (in Fourier case) or $[-1, 1]^2$ (in Chebyshev case) with their corresponding function values uniformly at random, to form the training and testing dataset, each of which has 100 data points. The loss function to be minimized is Mean Squared Error (MSE) between real function values and our model outputs. Adam optimizer with initial learning rate 0.01 (for $f_1$) or 0.05 (for $f_2, f_3$) and step scheduling strategy are adopted. The maximum training iterations is set to 80. Intuitive approximation results of numerical experiments are shown in Figure 2. The MSE obtained on test datasets for $f_1$, $f_2$, $f_3$ are $7.20 \times 10^{-4}$ ($n = 49$), $2.62 \times 10^{-4}$ ($n = 3$) and $4.51 \times 10^{-4}$ ($n = 6$) respectively.

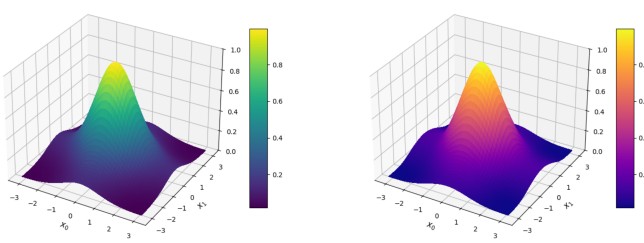

*(a)* Approximating $f_1(x_0, x_1)$ with Fourier series.

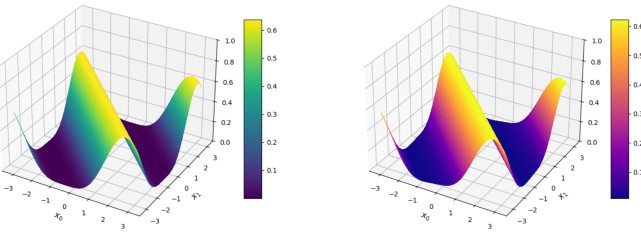

*(b)* Approximating $f_2(x_0, x_1)$ with Fourier series.

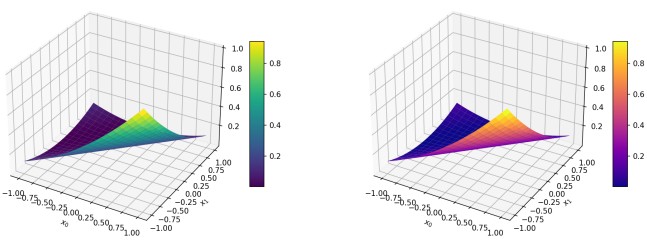

*(c)* Approximating $f_3(x_0, x_1)$ with Chebyshev series.

*Figure 2.* Numerical results of Fourier and Chebyshev series approximating target functions. The target functions are visualized in the left graphics and the right ones are learned from our model. Both of them are plotted with $50 \times 50$ grid samples of pre-defined intervals.

All of these experiments are conducted with qiskit (Javadi-Abhari et al., 2024), a python tool developed by IBM for quantum computing, on Intel(R) Xeon(R) Gold 5222 3.80GHz CPU. Parameter optimization is based on finite difference method, encapsulated in qiskit_algorithm package.

## 5. Summary

In this work, we have introduced the Spectral Adaptive Quantum Neural Network (SAQNN), a constructive quantum machine learning model with powerful expressivity. By leveraging a design inspired by LCU, SAQNN explicitly implements truncated Fourier and Chebyshev series on quantum circuits, offering a transparent and interpretable mechanism for function approximation. We have mathematically established its universal approximation property for multivariate square-integrable functions and demonstrated its spectral adaptability, which allows the model to has adjustable expressivity with layers and switch between function bases to effectively mitigate issues like the Gibbs phenomenon in aperiodic function approximation. A central contribution of our analysis is the derivation of rigorous error bounds for approximating Sobolev functions under $L_2$ norm. We proved that SAQNN achieves an optimal parameter complexity, matching the fundamental $n$-width lower bounds for both linear and nonlinear approximation methods. Moreover, our comparative analysis with SOTA of classical ReLU-FNNs highlights the quantum advantages in high-dimensional settings. While classical models often suffer from the curse of dimensionality regarding input dimension $d$, SAQNN maintains polynomial resource scaling, suggesting strong potential for quantum machine learning in complex, high-dimensional computational tasks.

It is worth mentioning that our model demonstrates a promising pathway for designing QNNs guided by quantum algorithms (e.g., LCU), which usually fully exploit quantum characteristics and may contribute to quantum advantage. Also, in SAQNN $\boldsymbol{\theta}$ are centralized, $\boldsymbol{x}$ and $\boldsymbol{\phi}$ in each layer are also separated. This modularity may enable advanced optimization techniques such as layer-wise freezing. Therefore, one can either develop novel architectures based on quantum algorithms or directly deploy SAQNN with suitable spectrum. Besides, SAQNN's modular structure can also help with analysis. For instance, to reduce circuit depth, we can trade expressivity for trainability by replacing state preparation block with a heuristic ansatz. Since this ansatz exclusively controls coefficient magnitudes (see Figure 1), it provides guidance for ansatz selection and simplifies circuit debugging.

Despite these theoretical advancements, several avenues for future research remain to enhance practical viability. While our model realizes quantum advantages, the circuit depth currently scales as $O(n \log n)$, presenting challenges for near-term quantum devices. Thus future works should fo-

cus on optimizing circuit synthesis to reduce depth without compromising expressivity. Additionally, analyses of the learning landscapes (such as the existence of barren plateau (McClean et al., 2018; Holmes et al., 2022; Mao et al., 2024), number of local minima (Bittel & Kliesch, 2021; Larocca et al., 2023), sample complexity (Huang et al., 2021; Cai et al., 2022)) of SAQNN are also interesting open problems. Furthermore, the numerical results in this paper only serves as the verification of feasibility of SAQNN, and we encourage the adaptation to various downstream machine learning tasks and examining the performance compared with the classical model, extending its utility beyond theoretical guarantees. For example, the continuous function outputs in SAQNN can be mapped to discrete class intervals and input features (pixels) can be encoded as $x$ for image classification.

## Acknowledgements

The authors thank Junhong Nie, Shuai Yang and Yunfei Yang for helpful discussions. This work was supported in part by the National Natural Science Foundation of China Grants No. 62325210, 12447107.

## Impact Statement

This paper presents work whose goal is to advance the field of Machine Learning. There are many potential societal consequences of our work, none of which we feel must be specifically highlighted here.

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

## A. Proof of Theorem 1

**Lemma 6** ((Sun et al., 2023)). *State preparation block of SAQNN can prepare any state with real amplitudes.*

**Theorem 1.** *For any multivariate square-integrable function $f : [-\pi, \pi]^d \to [0, 1]$ and any accuracy $\epsilon$, there exists a SAQNN $U_{\boldsymbol{\theta}, \boldsymbol{\phi}}(\boldsymbol{x})$ with parameters $\boldsymbol{\theta}$, $\boldsymbol{\phi}$ and rescale coefficient $a$ s.t.*

$$\|a \langle 0| U_{\boldsymbol{\theta}, \boldsymbol{\phi}}(\boldsymbol{x})^{\dagger} O U_{\boldsymbol{\theta}, \boldsymbol{\phi}}(\boldsymbol{x}) |0\rangle^{1/2} - f(\boldsymbol{x})\|_2 \leq \epsilon,$$

*where the global observable $O = |0\rangle \langle 0|$.*

*Proof.* Firstly, it is known by Lemma 4 that any arbitrary multivariant square-integrable function $f(\boldsymbol{x})$ can be $\epsilon$-approximated by a truncated Fourier series. Since $f : [-\pi, \pi]^d \to [0, 1]$ is square-integrable, for any $\epsilon > 0$, there exists $k \in \mathbb{N}^+$ and a finite series $f_k$ s.t.

$$\|f(\boldsymbol{x}) - f_k(\boldsymbol{x})\|_2 \leq \epsilon,$$

where

$$f_k = \sum_{j_1, j_2, \dots, j_d} c_{\boldsymbol{j}} e^{i \boldsymbol{j} \boldsymbol{x}}.$$

Let the sum of the modulus of the coefficients of $f_k$ be $\sum_{j_1, j_2, \dots, j_m} |c_{\boldsymbol{j}}| = a \geq 0$. Also, we have $n = (2k + 1)^d$, thus the number of target qubits of $P(\boldsymbol{\theta})$ is $m = \lceil d \log(2k + 1) \rceil$ in SAQNN.

Next, we prove the inequality in the theorem. For readability we change the index of series to $r$. That is,

$$f_k = \sum_{r=1}^{n} c_r e^{i \boldsymbol{j}^{(r)} \boldsymbol{x}}.$$

Based on section 3 and Lemma 6, the state preparation block can prepare any arbitrary state with real amplitudes. For any $a_r \in \mathbb{R}$ and $\sum_r a_r^2 = 1$, there exists $\boldsymbol{\theta}$ s.t.

$$P(\boldsymbol{\theta}) |0\rangle^m = \sum_{r=0}^{2^m - 1} a_r |r\rangle.$$

We can pad the remainder of series with 0s and use multi-qubit Toffoli gate before inversion of state preparation. The multi-qubit padding layer is under condition of top $m$ qubits being $|0\rangle$. For conditions of $|r\rangle$ that $r > n$, no controlled gate is inserted into the circuit and we can view them as some controlled-$I_2$ gates in the derivation. To be specific, denote $V_{\boldsymbol{\phi}}(\boldsymbol{x})$ as the matrix with $n$ layers of spectrum selection block with phase injection and padding layer, then

$$V_{\boldsymbol{\phi}}(\boldsymbol{x}) |r\rangle |\varphi\rangle = |r\rangle (e^{\phi_r} V_r |\varphi\rangle),$$

for $1 \leq r \leq n$,

$$V_{\boldsymbol{\phi}}(\boldsymbol{x}) |0\rangle |\varphi\rangle = |0\rangle (X \otimes I_{2^{d-1}} |\varphi\rangle),$$

and

$$V_{\boldsymbol{\phi}}(\boldsymbol{x}) |r\rangle |\varphi\rangle = |r\rangle (I_{2^d} |\varphi\rangle),$$

for $n + 1 \leq r \leq 2^m - 1$.

Then

$$\langle 0| U_{\boldsymbol{\theta},\boldsymbol{\phi}}(\boldsymbol{x}) |0\rangle = \langle 0| P(\boldsymbol{\theta})^{\dagger} V_{\boldsymbol{\phi}}(\boldsymbol{x}) P(\boldsymbol{\theta}) |0\rangle$$

$$= \langle 0| P(\boldsymbol{\theta})^{\dagger} V_{\boldsymbol{\phi}}(\boldsymbol{x}) ( \sum_{r=0}^{2^m-1} a_r |r\rangle) |0\rangle$$

$$= \langle 0| P(\boldsymbol{\theta})^{\dagger} (a_0 |0\rangle (X \otimes I_{2^{d-1}}) |0\rangle + \sum_{r=1}^{n} a_r |r\rangle e^{\phi_r} U_r(\boldsymbol{x}) |0\rangle + \sum_{r=n+1}^{2^m-1} a_r |r\rangle |0\rangle)$$

$$= \langle 0| ( \sum_{r=0}^{2^m-1} a_r^{\dagger} \langle r|)(a_0 |0\rangle (X \otimes I_{2^{d-1}}) |0\rangle + \sum_{r=1}^{n} a_r |r\rangle e^{\phi_r} U_r(\boldsymbol{x}) |0\rangle + \sum_{r=n+1}^{2^m-1} a_r |r\rangle |0\rangle)$$

$$= |a_0|^2 \langle 0| X \otimes I_{2^{d-1}} |0\rangle + \sum_{r=1}^{n} |a_r|^2 e^{\phi_r} \langle 0| U_r(\boldsymbol{x}) |0\rangle + \sum_{r=n+1}^{2^m-1} |a_r|^2 \langle 0| I_{2^d} |0\rangle$$

$$= \sum_{r=1}^{n} |a_r|^2 e^{\phi_r} \langle 0| U_r(\boldsymbol{x}) |0\rangle + \sum_{r=n+1}^{2^m-1} |a_r|^2$$

$$= \sum_{r=1}^{n} |a_r|^2 e^{\phi_r} e^{i \boldsymbol{j}^{(r)} \boldsymbol{x}} + \sum_{r=n+1}^{2^m-1} |a_r|^2$$

For arbitrariness of state preparation block, we can assign $|a_r|^2 = |c_r|/a$ and $\phi_r = \arg c_r$ for $0 \le r \le n-1$ and $|a_r| = 0$ for $r \ge n$, satisfying $\sum_{r=0}^{2^m-1} |a_r|^2 = 1$. Then we have

$$\langle 0| U_{\boldsymbol{\theta},\boldsymbol{\phi}}(\boldsymbol{x}) |0\rangle = \sum_r |a_r|^2 e^{\phi_r} e^{i \boldsymbol{j}^{(r)} \boldsymbol{x}} = f_k/a = \sum_r c_r e^{i \boldsymbol{j}^{(r)} \boldsymbol{x}}/a.$$

It holds that

$$\langle 0| U_{\boldsymbol{\theta},\boldsymbol{\phi}}(\boldsymbol{x})^{\dagger} O U_{\boldsymbol{\theta},\boldsymbol{\phi}}(\boldsymbol{x}) |0\rangle^{1/2} = (\langle 0| U_{\boldsymbol{\theta},\boldsymbol{\phi}}(\boldsymbol{x})^{\dagger} |0\rangle \langle 0| U_{\boldsymbol{\theta},\boldsymbol{\phi}}(\boldsymbol{x}) |0\rangle)^{1/2} = |f_k|/a.$$

So there exists $\boldsymbol{\theta}$ (which yields arbitrary $a_r$), $\boldsymbol{\phi}$ and rescale coefficient $a > 0$ s.t.

$$\|a \langle 0| U_{\boldsymbol{\theta},\boldsymbol{\phi}}(\boldsymbol{x})^{\dagger} O U_{\boldsymbol{\theta},\boldsymbol{\phi}}(\boldsymbol{x}) |0\rangle^{1/2} - f(\boldsymbol{x})\|_2 = \||f_k| - f\|_2 \le \|f_k - f\|_2 \le \epsilon.$$

$\square$

Remark that if $a$ does not match the sum of the modulus of the coefficients of $f_k$, we can set $|a_0|^2 = 1 - \sum_{r=1}^{n}(|c_r|/a)$. It works as a regulator and allows any $a \ge \sum_{r=1}^{n} |c_r|$ without affecting the correctness of Theorem 1.

## B. Fine-tuning Strategy of the Rescale Coefficient

The rescale coefficient $a$ in Theorem 1 works as a role to break the normalization condition for coefficients of the state prepared, which corresponds to the coefficients in Fourier series. It greatly expands the expressivity of the model but $a$ is usually unknown in machine learning tasks. However, the sum of modulus of coefficients of a truncated Fourier series can be bounded by number of items $n$. Firstly we have

$$\sum_{\boldsymbol{j}} |c_{\boldsymbol{j}}| = \sum_{\boldsymbol{j}} \frac{1}{(2\pi)^d} \int_{[-\pi,\pi]^d} f(\boldsymbol{x}) e^{-i\boldsymbol{j}\boldsymbol{x}} \le \sum_{\boldsymbol{j}} \frac{1}{(2\pi)^d} \int_{[-\pi,\pi]^d} |f(\boldsymbol{x}) e^{-i\boldsymbol{j}\boldsymbol{x}}| \le \sum_{\boldsymbol{j}} \frac{1}{(2\pi)^d} \int_{[-\pi,\pi]^d} |f(\boldsymbol{x})||e^{-i\boldsymbol{j}\boldsymbol{x}}|.$$

Since $|f(\boldsymbol{x})| \le 1$ and $|e^{-i\boldsymbol{j}\boldsymbol{x}}| \le 1$,

$$\sum_{\boldsymbol{j}} |c_{\boldsymbol{j}}| \le \sum_{\boldsymbol{j}} \frac{1}{(2\pi)^d} \int_{[-\pi,\pi]^d} 1 = \sum_{\boldsymbol{j}} \frac{1}{(2\pi)^d} (2\pi)^d = n.$$

Denoting the re-scale coefficient to be fine-tuned as $a'$. In proof of Theorem 1 we can see that our model without rescaling can express arbitrary Fourier series with sum of modulus of coefficients less than or equal to 1. So it still holds when we set $a'$ to its upper bound, but on real-world devices the numerical precision will be insufficient and the learning landscape is poor when $a$ is actually far away from the upper bound, which is common. To make our model practical, we further give a fine-tuning strategy to find the appropriate $a'$. Our strategy consists of two steps:

1. Normalize the dataset, remapping the value to interval [0,1] with setting the max value to 1,

2. Start from $a' = 1$, repeat multiplying $a'$ by 2 until it learns well.

In our strategy, the first step ensures that $\max_{\boldsymbol{x}} f(\boldsymbol{x}) = 1$. And we have a theoretical upper bound for $f_k$:

$$f_k(\boldsymbol{x}) = \sum_{\boldsymbol{j}}^{\|\boldsymbol{j}\|_\infty \leq k} c_{\boldsymbol{j}} e^{i\boldsymbol{j}\boldsymbol{x}} \leq \sum_{\boldsymbol{j}}^{\|\boldsymbol{j}\|_\infty \leq k} |c_{\boldsymbol{j}} e^{i\boldsymbol{j}\boldsymbol{x}}| \leq \sum_{\boldsymbol{j}}^{\|\boldsymbol{j}\|_\infty \leq k} |c_{\boldsymbol{j}}||e^{i\boldsymbol{j}\boldsymbol{x}}| \leq \sum_{\boldsymbol{j}}^{\|\boldsymbol{j}\|_\infty \leq k} |c_{\boldsymbol{j}}| = a.$$

While the $L_2$-approximation $f_k$ might strictly have a peak slightly less than 1, the constraint $a < 1$ would impose an artificial hard ceiling on the model. This would render the model mathematically incapable of representing any function values in the range $(a, 1]$. To avoid limiting the model's expressivity and to ensure it has the capacity to approximate the normalized target functions, we must permit $a \geq 1$. Therefore, starting the fine-tuning search from $a' = 1$ is the minimal requirement to ensure the hypothesis space covers the target function's scale, which is set as the starting point in step 2.

With step 2, we can see the training loss would be high unless it exceeds the real value for the first time. Right after that, at least we have $a' \geq a/2$, which means the amplitudes we try to learn for series have sufficient proportion.

## C. Switch to Chebyshev Series and Model Modifications

Fourier series are effective for periodic functions but struggle with aperiodic data. We resolve this by introducing the Chebyshev series (Mason & Handscomb, 2002), creating a complementary framework where each kind of function basis adapts to different approximation needs. This appendix outlines how our model implements Chebyshev series through simple modifications.

The Chebyshev series of a univariant function $f(x) \in L^2([-1, 1])$ is the linear combination of Chebyshev polynomials,

$$f(x) = \sum_{j=0}^{\infty} b_j T_j(x),$$

where $T_j(x)$ satisfies the recursion relations

$$T_0(x) = 1,$$
$$T_1(x) = x,$$
$$T_{j+1}(x) = 2x T_j(x) - T_{j-1}(x).$$

The closed form of $T_j(x)$ is

$$T_j(x) = \cos\left(j \arccos x\right).$$

The Chebyshev polynomials of multivariant function $f(\boldsymbol{x}) \in L^2([-1, 1]^d)$ is built by tensor products of univariant Chebyshev polynomials,

$$f(\boldsymbol{x}) = \sum_{\boldsymbol{j}=0}^{\infty} b_{\boldsymbol{j}} T_{j_1}(x_1) T_{j_2}(x_2)...T_{j_d}(x_d).$$

Similar to the results of Fourier series, any square-integrable multivariant function $f(\boldsymbol{x})$ can be approximated by truncated Chebyshev series to arbitrary accuracy. A $k$-truncated Chebyshev series is defined as

$$f_k(\boldsymbol{x}) = \sum_{\boldsymbol{j}=\boldsymbol{0}}^{\|\boldsymbol{j}\|_\infty \leq k} b_{\boldsymbol{j}} T_{j_1}(x_1) T_{j_2}(x_2)...T_{j_d}(x_d).$$

**Lemma 7.** *(Chebyshev approximation ([Mason & Handscomb, 2002; Canuto et al., 2006](#))) For any $f \in L_2([-1,1]^d)$ and $\epsilon > 0$, there exists a $k$-truncated Chebyshev series $f_k$ s.t.*

$$\|f(\boldsymbol{x}) - f_k(\boldsymbol{x})\|_2 \le \epsilon.$$

To implement Chebyshev series, we replace each $R_z$ gate in spectrum selection block with $R_y$ gate. For example, to implement $T_{j_1^{(r)}}(x_1)$, we use

$$R_y(2j_1^{(r)} \arccos x_1) = \begin{pmatrix} T_{j_1^{(r)}}(x_1) & * \\ * & * \end{pmatrix},$$

satisfying

$$\langle 0 | U_r(\boldsymbol{x}) | 0 \rangle = T_{j_1^{(r)}}(x_1) T_{j_2^{(r)}}(x_2) ... T_{j_d^{(r)}}(x_d).$$

Here we also change the pre-process of inputs compared to that in implementation of Fourier series, still getting rid of classical trainable weights.

Another difference is that the coefficients of Chebyshev series are real numbers. To make it suitable for Chebyshev case, the phase injection degenerates into the case whether $\phi_i = 0$ or $\phi_i = \pi$, corresponding to the 1 or -1 multiplied on each Chebyshev polynomial. With these changes to our model, we can prove UAP of SAQNN based on Chebyshev series in the completely similar way of Theorem 1, details omitted here.

## D. Proof of Theorem 2

We go further to show the quantum resource needed to obtain UAP. Since Sobolev spaces and Fourier series have been considerably studied, we first introduce a cutting-edge lemma adapted from approximation theory and then give a cost analysis for our model.

**Definition 8.** (Approximation numbers ([Kühn et al., 2016](#))) The $n$-th approximation number of bounded linear operator $T : X \to Y$ between two Banach spaces is

$$a_n(T : X \to Y) := \inf_{A:X \to Y} \{\|T - A\| : \text{rank } A \le n\}.$$

Approximation numbers describe accuracy of independent $n$ terms to uniformly approximate the whole function space under certain mapping relations. Since we consider the optimal linear $L_2$-approximation by Sobolev functions of finite rank, we focus on the map id : $H_u^s([-\pi,\pi]^d) \to L_2([-\pi,\pi]^d)$, which is called Sobolev embeddings.

**Lemma 9** (Error bounds of Fourier series approximating Sobolev functions ([Kühn et al., 2016](#))). *For $s > 0$, we have*

$$a_n(id : H_u^s([-\pi,\pi]^d) \to L_2([-\pi,\pi]^d)) \le \begin{cases} 4^s \left( \frac{\log(1 + d/\log n)}{\log n} \right)^{s/2} & : d \le n \le 2^d \\ 4^s d^{-s/2} n^{-s/d} & : n \ge 2^d, \end{cases}$$

*$a_n$ is reached by minimal $n$ terms of Fourier series regarding with $\|\boldsymbol{j}\|_2$ and $A$ is the projection of id : $H_u^s([-\pi,\pi]^d) \to L_2([-\pi,\pi]^d)$ on the space spanned by optimal Fourier series.*

Here we neglect the case of $n < d$ in which the approximation to arbitrary accuracy is impossible and $a_n$ is trivial.

**Theorem 2.** *For any multivariate Sobolev function $f \in H_u^s([-\pi,\pi]^d)$ with value range $[0,1]$ and any $\epsilon > 0$, there exists a SAQNN $U_{\boldsymbol{\theta},\boldsymbol{\phi}}(\boldsymbol{x})$ with parameters $\boldsymbol{\theta}$, $\boldsymbol{\phi}$ and rescale coefficient $a$ s.t.*

$$\|a \langle 0 | U_{\boldsymbol{\theta},\boldsymbol{\phi}}(\boldsymbol{x})^{\dagger} O U_{\boldsymbol{\theta},\boldsymbol{\phi}}(\boldsymbol{x}) | 0 \rangle^{1/2} - f(\boldsymbol{x})\|_2 \le \epsilon,$$

*where the global observable $O = |0\rangle \langle 0|$. The circuit width (number of qubits) of $U_{\boldsymbol{\theta},\boldsymbol{\phi}}(\boldsymbol{x})$ is $O(\log n)$, the circuit depth is $O(n \log n)$ and the parameter complexity is $O(n)$. Here $n = O((d + 1/\epsilon)^{16(1/\epsilon)^{2/s}})$.*

*Proof.* It's known by definition that $f \in H_u^s([-\pi,\pi]^d) \subset L_2([-\pi,\pi]^d)$ for $s > 0$, so by Lemma 4 that that exists $k \in \mathbb{N}^+$ and a finite Fourier series $f_k$ s.t.

$$\|f_k(\boldsymbol{x}) - f(\boldsymbol{x})\|_2 \le \epsilon.$$

Let the sum of the modulus of coefficients of $f_k$ be $a$, then in a similar way of Theorem 1 there exists $\boldsymbol{\theta}, \boldsymbol{\phi}$ s.t.

$$\langle 0| U_{\boldsymbol{\theta},\boldsymbol{\phi}}(\boldsymbol{x})^\dagger O U_{\boldsymbol{\theta},\boldsymbol{\phi}}(\boldsymbol{x}) |0\rangle = \langle 0| U_{\boldsymbol{\theta},\boldsymbol{\phi}}(\boldsymbol{x})^\dagger |0\rangle \langle 0| U_{\boldsymbol{\theta},\boldsymbol{\phi}}(\boldsymbol{x}) |0\rangle = f_k^2/a^2.$$

Then

$$\|a \langle 0| U_{\boldsymbol{\theta},\boldsymbol{\phi}}(\boldsymbol{x})^\dagger O U_{\boldsymbol{\theta},\boldsymbol{\phi}}(\boldsymbol{x}) |0\rangle^{1/2} - f(\boldsymbol{x})\|_2 \le \|f_k(\boldsymbol{x}) - f(\boldsymbol{x})\|_2 \le \epsilon$$

Next we analyze the circuit cost. By Lemma 9, we have

$$4^s d^{-s/2} \le a_n \le 4^s \left( \frac{\log(1 + d/\log d)}{\log d} \right)^{s/2} \tag{1}$$

when $d \le n \le 2^d$, and

$$0 < a_n \le 2^s d^{-s/2}$$

when $n \ge 2^d$.

Case 1. $4^s \left( \frac{\log(1+d/\log d)}{\log d} \right)^{s/2} \le \epsilon \le 1$. Then it holds for all $n \ge d$ that

$$a_n \le 4^s \left( \frac{\log(1 + d/\log d)}{\log d} \right)^{s/2} \le \epsilon.$$

So we only need to set $n = d$ to get accuracy $\epsilon$.

Case 2. $4^s d^{-s/2} \le \epsilon \le 4^s \left( \frac{\log(1+d/\log d)}{\log d} \right)^{s/2}$. To ensure the accuracy $\epsilon$, the upper bound of $a_n$ should satisfies

$$4^s \left( \frac{\log(1 + d/\log n)}{\log n} \right)^{s/2} \le \epsilon.$$

For the convenience of solving $n$, we relax the upper bound of $a_n$,

$$\frac{\log(1 + d/\log n)}{\log n} \le \frac{\log(1 + d/\log d)}{\log n} \le \frac{\log(1 + d)}{\log n},$$

where $d \ge 2$. Then solve $n$ from the inequality

$$4^s \left( \frac{\log(1 + d)}{\log n} \right)^{s/2} \le \epsilon,$$

we have

$$\log n \ge 16(1/\epsilon)^{2/s} \log(1 + d),$$

which implies

$$n \ge (d + 1)^{16(1/\epsilon)^{2/s}}.$$

Case 3. $2^s d^{-s/2} \le \epsilon \le 4^s d^{-s/2}$. Recall that $a_n \le 2^s d^{-s/2}$ for $n \ge 2^d$, we only need to set $n = 2^d$.

Case 4. $0 \le \epsilon \le 2^s d^{-s/2}$. In this case, we solve the inequality

$$4^s d^{-s/2} n^{-s/d} \le \epsilon,$$

then we get

$$n \ge 4^d (1/\epsilon)^{d/s} d^{-d/2}.$$

So

$$\log n \ge 2d + \frac{d}{s} \log(1/\epsilon) - \frac{d}{2} \log d.$$

Relaxing $\log d$ to 1, we have

$$\log n \geq \frac{3}{2}d + \frac{d}{s}\log(1/\epsilon).$$

Since $\epsilon \leq 2^s d^{-s/2}$, we have an upper bound $d \leq 4(1/\epsilon)^{2/s}$. Then

$$\log n \geq 6(1/\epsilon)^{2/s} + \frac{4}{s}(1/\epsilon)^{2/s}\log(1/\epsilon),$$

which implies

$$n \geq 2^{6(1/\epsilon)^{2/s}}(1/\epsilon)^{\frac{4}{s}(1/\epsilon)^{2/s}}.$$

To rule them all in four cases, we give an inequality that holds for any $0 \leq \epsilon \leq 1$ and $d \geq 2$:

$$n \geq (d + 1/\epsilon)^{16(1/\epsilon)^{2/s}}.$$

Due to the optimality of Fourier series approximating Sobolev spaces under $L_2$ norm (Pinkus, 2012), the minimal $n$ to achieve accuracy $\epsilon$ is $(d + 1/\epsilon)^{16(1/\epsilon)^{2/s}}$.

Back to our model, to realize a Fourier series with $n$ terms, the state preparation block should prepare $O(n)$ different real amplitudes, and the phase injection block should pair each amplitude with a phase, which also produces $O(n)$ parameters. So the parameter complexity of our model is totally $O(n) = O((d + 1/\epsilon)^{16(1/\epsilon)^{2/s}})$.

Further, we need $O(\log n)$ qubits of state preparation block to generate $O(n)$ amplitudes, and extra $O(d)$ qubits to apply $U_r$. However, we can use only one $R_z$ gate to implement each $U_r$,

$$U_r(\boldsymbol{x}) = R_z(-2\boldsymbol{j}^{(r)}\boldsymbol{x}),$$

which still satisfies

$$\langle 0| U_r(\boldsymbol{x}) |0\rangle = e^{i\boldsymbol{j}^{(r)}\boldsymbol{x}}.$$

The inputs $\boldsymbol{x}$ are more densely encoded for reducing qubit cost of SAQNN.

This optimization technique uses a different pre-process to compresses inputs into one rotation gate while not affecting the correctness of Theorem 1 and Theorem 2. So the number of qubits needed here can be reduced to 1. The width (number of qubits) of our model is $O(\log n) = (1/\epsilon)^{2/s}\log(d + 1/\epsilon)$.

To analyze the circuit depth, we should take use of Lemma 3, which implies a $O(2^{k-1})$ depth synthesis for $k$-qubit multiplexor-$R_y$ gate. So the circuit depth of state preparation block is $\sum_{k=1}^{m} 2^{k-1} = O(2^m) = O(n)$. In spectrum selection block, we have $O(n)$ layers. In each layer, we can synthesize the controlled-$R_z$ gate as below:

It's known that each $(m+1)$-qubit Toffoli gate can be implemented by CNOT gate in depth $O(m)$ (Saeedi & Pedram, 2013). So the circuit depth of spectrum selection block is $O(mn) = O(n\log n)$. Similarly the depth of inversion of state preparation block is also $O(n)$. As a whole, the circuit depth of our model is $O(n\log n) = O((d+1/\epsilon)^{16(1/\epsilon)^{2/s}}(1/\epsilon)^{2/s}\log(d + 1/\epsilon))$.

$\square$

## E. Parameter Optimality

In approximation theory, the approximation methods mainly conclude linear methods (Pinkus, 2012) and nonlinear methods (DeVore, 1998). Linear methods refer to the linear combination of simple function basis, such as Fourier series, Chebyshev series, Bernstein polynomials and so on. They span linear subspaces to approximate target functions and takes the form

$$A(f) = a_0 f_0 + a_1 f_1 + ... + a_l f_l.$$

Nonlinear approximation methods remove the constraint of linear dependence on parameters. The approximating function is a nonlinear function of the parameters, or the selection of basis functions itself depends on the function being approximated. The approximation $A(f)$ does not have a simple linear structure. A well-known example for nonlinear methods is FNNs with single hidden layer, whose expression can be written as

$$A(f) = \sigma(\sum_i w_i x_i),$$

where $\sigma$ is the nonlinear activation function. It has been proved to have UAP in early years (Cybenko, 1989; Hornik et al., 1989).

When considering the number of parameters needed to approximate specific functions, the commonly used indicators are Kolmogorov $n$-width for linear methods and manifold $n$-width for nonlinear methods.

**Definition 10** (Kolmogorov $n$-width (Kolmogoroff, 1936; Pinkus, 2012))**.** Let $K$ be a subset of a normed linear space $X$. The Kolmogorov $n$-width $d_n(K, X)$ measures the error of the best approximation to $K$ by $n$-dimensional linear subspaces of $X$. It is defined as:

$$d_n(K, X) := \inf_{X_n} \sup_{f \in K} \inf_{g \in X_n} \|f - g\|_X,$$

where the outer infimum is taken over all linear subspaces $X_n \subset X$ of dimension at most $n$.

This quantity characterizes the fundamental limit of linear approximation methods. Note that Kolmogorov $n$-width $d_n(H^s([-\pi, \pi]^d), L^2([-\pi, \pi]^d))$ is same to the $n$-th approximation number $a_n(H^s([-\pi, \pi]^d) \to L^2([-\pi, \pi]^d))$ in our case.

To characterize the approximation power of nonlinear parameterized models, we consider the Manifold $n$-width.

**Definition 11** (Manifold $n$-width (DeVore et al., 1989; Maiorov & Ratsaby, 1999))**.** Let $M_n$ denote a continuous manifold in $X$ parameterized by $n$ real variables, known as $M_n = \{R(a) : a \in \mathbb{R}^n\}$ for a continuous mapping $R : \mathbb{R}^n \to X$. The Manifold $n$-width $\delta_n(K, X)$ is defined as

$$\delta_n(K, X) := \inf_{M_n} \sup_{f \in K} \inf_{g \in M_n} \|f - g\|_X.$$

It has been shown in approximation theory that in terms of the scaling of parameter count $n$, it satisfies (DeVore et al., 1989; Edmunds & Triebel, 1996; Pinkus, 2012; Geller & Pesenson, 2014)

$$d_n(H^s([-\pi, \pi]^d), L^2([-\pi, \pi]^d)) \asymp \delta_n(H^s([-\pi, \pi]^d), L^2([-\pi, \pi]^d)) \asymp n^{-s/d},$$

here $a \asymp b$ means that there exists constants $c_1, c_2$ s.t. $c_1 b \leq a \leq c_2 b$.

The parameter efficiency of linear methods is same as that of nonlinear methods. So the $O(n)$ parameter complexity of SAQNN is optimal among linear and nonlinear methods, with all quantum and classical neural networks not being better than our model in asymptotic order of parameters when approximating Sobolev functions to accuracy $\epsilon$ under $L_2$ norm.

