# OpenReview forum: "SAQNN: Spectral Adaptive Quantum Neural Network as a Universal Approximator"
_ICML.cc/2026/Conference — ICML 2026 regular_

### Official Review · Reviewer_hY5x · 2026-02-18

**Soundness:** 3
**Presentation:** 3
**Significance:** 3
**Originality:** 3
**Overall Recommendation:** 4
**Confidence:** 3

**Summary:**

The paper introduces the Spectral Adaptive Quantum Neural Network, which is a circuit architecture for quantum machine learning. They demonstrate that the architecture possesses the universal approximation property. After providing the notation, the scheme, the theorems, and error bounds, they show some numerical experiments.

**Compliance With Llm Reviewing Policy:**

Affirmed.

**Final Justification:**

I think the paper is on the borderline of acceptance. Whereas the research is solid, my concerns regarding the practical applications beyond academia remain. Ultimately, it is the chair's decision whether this type of work aligns with the conference's scope.

**Key Questions For Authors:**

Can one simulate an example on a more realistic machine-learning task? How does the influence of Barren plateaus impact the usefulness of the algorithm?

**Limitations:**

To some degree, but the authors could further discuss both the technical challenges of the quantum machine learning field as a whole and the problems arising through Barren plateaus latter is briefly mentioned in the conclusion.

**Strengths And Weaknesses:**

Soundness: The submission is technically sound.
Presentation: The presentation is good.
Significance: Whereas the authors address an interesting problem, I question whether or not this manuscript is suitable for ICML, as it lacks any significant empirical results. Furthermore, its practical relevance (quantum machine learning, that is, currently or in the future) for the general machine learning community is highly questionable.
Originality: The paper is original and deepens our understanding of quantum machine learning.

---

> ### Author Rebuttal · Authors · 2026-03-30
>
> We thank the reviewer for raising the issues worthy of discussion. We are willing to make responses in the following.
>
> > 1. Whereas the authors address an interesting problem, I question whether or not this manuscript is suitable for ICML, as it lacks any significant empirical results.
> >
> >    Can one simulate an example on a more realistic machine-learning task?
>
> Reply 1: We believe that the theory for learning models is also a valued topic in ICML. Moreover, given nascent stages of QML, theoretical research is crucial for long-term development of this field. To clarify, the primary contribution of our manuscript is mainly theoretical, positioned as foundational research exploring the approximation capabilities of QNNs. Rather than targeting large-scale empirical benchmarks, our numerical experiments were designed as proof-of-concept validations to better illustrate and verify our proposed theorems. Specific reasons for the experiment design are detailed below:
>
> - Our theoretical results (Theorems 1 and 2) establish UAP properties for approximating functions within Sobolev spaces. Practical machine learning datasets consist of discrete data points without a well-defined target function. Therefore, it is more rigorous and intuitive to use bivariate functions instead of "datasets" to validate our theory of "function" approximation.
>
> - Real-world datasets typically possess large dimensions. Simulating quantum circuits leads to an exponential overhead in classical computing resources. Also, current quantum computing devices are not yet mature enough to support large-scale, practical machine learning applications. However, with the rapid development of quantum technology, we expect that models like SAQNN will be effectively adapted to solve real-world machine learning tasks in the foreseeable future.
>
> > 2. Furthermore, its practical relevance (quantum machine learning, that is, currently or in the future) for the general machine learning community is highly questionable.
>
> We appreciate the reviewer's candid perspective. We agree that the practical relevance of QML is currently a subject of debate within the broader ML community. However, we respectfully argue that it's exactly the ongoing controversy that makes theoretical investigations essential.
>
> The skepticism surrounding QML generally stems from two core questions: (1) Does it truly possess superior expressive power compared to mature classical models? and (2) Can it be practically deployed on real-world tasks? Our work partially answers the first question by proving that SAQNN achieves advantages over classical FNNs. At the current stage, exploring and quantifying the expressive power and resource boundaries of quantum models theoretically can provide useful references for future model design. For the long-term development of the field as a whole, foundational exploration is significant and highly worth attempting.
>
> Regarding the second question, while current near-term hardware limits immediate large-scale deployment, the quantum computing industry is advancing at an unprecedented pace. Historically, in classical deep learning, foundational approximation theories were established well before the hardware (e.g., modern GPUs) and software frameworks existed to make them ubiquitous. By providing a constructive, mathematically guaranteed blueprint today, we are laying the necessary theoretical groundwork. When fault-tolerant quantum hardware matures, the general ML community will have reliable, theory-backed quantum architectures ready for practical use, rather than relying on uninterpretable empirical heuristics.
>
> > 3. How does the influence of Barren plateaus impact the usefulness of the algorithm?
>
> In QML, Barren plateaus (BP) occurs when the variance of the gradients vanishes exponentially, causing the training landscape to become overwhelmingly flat and making optimization practically impossible.
>
> We sincerely refer the reviewer to **Reply 1** of our rebuttal to **Reviewer Sekk** for our preliminary observations and results of training landscape. Notably, while the vanishing gradient problem in classical deep neural networks is primarily induced by network depth (the number of layers), the Barren Plateau (BP) phenomenon in quantum neural networks is heavily influenced by network width (the number of qubits). As proven in Theorem 2, SAQNN only needs a circuit width of only $O(\log n)$. We conjecture that this property theoretically shields SAQNN from the BP effects, though rigorous analytical calculations and proofs are still required. Systematically studying the BP phenomenon and the training dynamics of SAQNN at scale is a primary focus for our next phase of research, which we will vigorously pursue.

---

> > ### Author Rebuttal · Reviewer_hY5x · 2026-04-01
> >
> > Thank you for addressing my points. I updated my recommendation to a weak accept because the authors present compelling arguments. However, I still think there is a need for a better understanding of how Barren plateaus affect real-world applications, to conclusively judge the impact of their results. Therefore, I do not fully accept.

---

### Official Review · Reviewer_Sekk · 2026-03-11

**Soundness:** 3
**Presentation:** 4
**Significance:** 4
**Originality:** 4
**Overall Recommendation:** 5
**Confidence:** 3

**Summary:**

In this paper, the authors introduce the Spectral Adaptive Quantum Neural Network (SAQNN), a constructive quantum machine learning architecture designed to serve as a universal approximator. Authors prove that SAQNN possesses the Universal Approximation Property for any multivariate square-integrable function. Moreover, they establish non-asymptotic error bounds for approximating Sobolev functions under the L2 norm, demonstrating that their model achieves optimal parameter complexity. Last, they show that SAQNN overcomes the curse of dimensionality.

**Compliance With Llm Reviewing Policy:**

Affirmed.

**Final Justification:**

Thank you to the authors for their comprehensive response and the additional context provided during the rebuttal phase. By directly addressing my concerns about the baselines, they have further highlighted the originality and significance of this work. This detailed response reinforces my prior evaluation.

**Key Questions For Authors:**

1-In real-world datasets, the input dimension d can be massive. While your theoretical bounds hold, did you observe any overheads or optimization difficulties when simulating dimensions larger than d=2?
2-Could you elaborate on how coefficient a's scale might impact the gradients during the actual Adam optimization process?

**Limitations:**

The authors have adequately discussed the technical limitations of their work, specifically noting the challenging O(nlogn) circuit depth for near-term hardware and the need to investigate learning landscapes in future work. However, they dismiss the broader societal impact.

**Strengths And Weaknesses:**

For soundness, I believe that theoretical foundation of this paper is highly rigorous. By leveraging established approximation theory, specifically Fourier and Chebyshev, authors provide solid mathematical proofs for their claims regarding UAP and error bounds. However, it would be better to see experiments with different dimensions, not limited with two, and training points for observing how model behaves under various circumstances.
For presentation, I think manuscript is well-written and easy to follow.
For significance, as a researcher who works with high dimensional datasets, I find the theoretical circumvention of the curse of dimensionality to be a highly significant contribution. The fact that the model's costs remain polynomial relative to the input dimension d suggests immense long-term utility for processing complex social or behavioral data. Last, with implementing Fourier and Chebyshev techniques, it also incorporates an interpretable framework for machine learning.
For originality, I believe that authors offer valuable perspective to the field. Instead of relying on heuristic approaches, they introduce a constructive model that naturally embeds classical approximation theory.

---

> ### Author Rebuttal · Authors · 2026-03-30
>
> We appreciate the reviewer for recognition of the contributions and novelty of our work! We have carefully considered your feedback and provide our replies below.
>
> > 1. In real-world datasets, the input dimension d can be massive. While your theoretical bounds hold, did you observe any overheads or optimization difficulties when simulating dimensions larger than d=2?
>
> Reply 1: We thank the reviewer for raising these valuable questions. At first we clarify that our work does not incur any additional quantum overhead in high-dimensional cases and is aimed at theoretical analysis of QNN expressivity, design of optimization algorithms belonging to the future work on which we plan to conduct more specific and systematic research. However, we are willing to provide a detailed explanation of the additional costs associated with classical simulations and some preliminary observations and results of training landscape, shown as below.
>
> - Exponential overhead of classical simulation: We believe it is essential to distinguish between the theoretical execution cost on quantum hardware and the computational cost of simulating quantum circuits on classical hardware. Although Theorem 2 rigorously proves that SAQNN's quantum resources (width, depth, and parameters) scale polynomially with $d$, state-vector simulation of these highly entangled circuits (especially the multiplexor-$R_y$ and multi-controlled gates) on classical devices incurs an exponential time complexity. Even when utilizing high-performance simulators like Qiskit-Aer on remote servers, the classical simulation cost for $d>2$ quickly becomes prohibitive, with circuit execution times increasing rapidly.
>
> - Training landscape: We conducted supplementary simulations on functions $f'(x_0,x_1,x_2)=e^{-(\sin^2(x_0/2)+\sin^2(x_1/2)+\sin^2(x_2/2))}$ and $f''(x_0,x_1,x_2,x_3)=\frac{\cos(x_0+x_1+x_2+x_3+\pi/6)}{2.5}$, and observed the phenomenon of local minima. Specifically, the training loss sometimes decreases steadily during the initial stages but ultimately stagnates at a suboptimal value. However, we also found that this issue can be mitigated to some extent through simple improvements to the optimization algorithm, such as introducing random perturbations or applying learning rate annealing. Additionally, we did not observe any signs of barren plateaus when training functions $f'$ and $f''$, which we consider a highly positive signal. It is worth noting that, theoretically, while the vanishing gradient problem in classical deep neural networks is primarily induced by an increase in network depth (the number of layers), gradients in quantum neural networks typically decay exponentially with an increase in network width (the number of qubits). Therefore, this favorable training behavior is likely attributable to SAQNN's remarkably compact network width of $O(\log n)$.
>
> > 2. Could you elaborate on how coefficient a's scale might impact the gradients during the actual Adam optimization process?
>
> Reply 2: Assuming a Mean Squared Error (MSE) loss $L = \frac{1}{N}\sum (a\hat{y}\_q - y)^2$ (here $\hat{y}\_q$ is model output without scaling), the gradient with respect to a circuit parameter $\theta$ is $\nabla_\theta L = 2a^2(\hat{y}\_q - y/a)\nabla_\theta \hat{y}\_q$. The Hessian matrix is $H = 2a^2 \nabla_\theta\hat{y}\_q (\nabla_\theta\hat{y}\_q)^T + 2a^2(\hat{y}\_q - y/a)\nabla_\theta^2\hat{y}\_q$. This formulation reveals that a large $a$ not only drastically scales the overall gradient magnitude by $a^2$ but also effectively shifts the optimization target for the original output to a tiny value $y/a$. While the Adam optimizer's scale-invariance property attempts to normalize the gradient, the resulting change in the final scaled output is magnified by $a$ (i.e., $\Delta \hat{y} \approx a \cdot \nabla_\theta \hat{y}_q \cdot \eta$). A mismatch occurs that the QNN is forced to hit a highly precise target $y/a$, yet the optimizer dictates amplified strides ($a \cdot \eta$) in the output space, which may lead to overshooting, preventing the model from converging to the optimal point. This is one of the reasons why we adopt re-scale coefficient fine-tuning strategy starting from $a=1$.

---

> > ### Author Rebuttal · Reviewer_Sekk · 2026-04-04
> >
> > Thank you for the detailed and thoughtful rebuttal, which has fully addressed my questions and reinforced my support for this paper.

---

### Official Review · Reviewer_u459 · 2026-03-12

**Soundness:** 3
**Presentation:** 2
**Significance:** 3
**Originality:** 3
**Overall Recommendation:** 5
**Confidence:** 4

**Summary:**

This paper proposes a constructive quantum neural network model and proves that it has the universal approximation property, meaning it can approximate any square-integrable function to arbitrary accuracy. The authors further claim that the model supports switching function bases, making it adaptable across different approximation and learning settings. Beyond universality, the paper argues that the proposed construction has asymptotic advantages over the best classical feed-forward neural networks in circuit size and achieves optimal parameter complexity for approximating Sobolev functions.

**Compliance With Llm Reviewing Policy:**

Affirmed.

**Final Justification:**

This is a good work that suitable for this conference.

**Key Questions For Authors:**

- What are the main new insights of this work compared with existing universal approximation and expressivity results for QNNs?
- How can the proposed constructive model be translated into practical quantum machine learning algorithms, and how realistic are the claimed advantages under implementation constraints?

**Limitations:**

- The broader impact would be clearer if the paper better explained how these approximation results inform the design or analysis of practical QML models and algorithms.
- Some additional intuition about the model construction and why it leads to these approximation properties would improve accessibility.
- Better explain the insights with existing expressivity and universal approximation literature in QML.

**Strengths And Weaknesses:**

Strengths
- The paper studies an important and timely topic in quantum machine learning: the expressivity and approximation power of quantum neural networks.
- Overall, this appears to be a potentially significant theoretical contribution to QML foundations.

Weaknesses
- The core direction is strong, but some important aspects need clearer explanation. For exmaple, in Theorem 1, more details are needed for the QNN. Only existing claim is not strong.
- The L2 norm should be better justified. While this is a standard choice in approximation theory, the paper should explain why it is the most relevant notion here and what its limitations are for learning tasks.
- The paper would benefit from more discussion of the assumptions under which the asymptotic advantages hold.

---

> ### Author Rebuttal · Authors · 2026-03-30
>
> We appreciate the reviewer's detailed and helpful feedback. Here we respond to the insightful comments and questions below.
>
> > 1. The core direction is strong, but some important aspects need clearer explanation. For exmaple, in Theorem 1, more details are needed for the QNN. Only existing claim is not strong.
>
> Reply 1: Thanks for pointing out this issue. We will improve the explanation and carefully review and revise other unclear explanations in the manuscript.
>
> > 2. The L2 norm should be better justified. While this is a standard choice in approximation theory, the paper should explain why it is the most relevant notion here and what its limitations are for learning tasks.
>
> Reply 2: We chose the $L_2$ norm because it is directly related to the Mean Squared Error (MSE), which is a widely-used loss function in practical machine learning.
>
> As an average-case metric, the $L_2$ norm may under-penalize localized predictive errors (e.g., local spikes), which could affect point-wise reliability.
>
> > 3. The paper would benefit from more discussion of the assumptions under which the asymptotic advantages hold.
>
> Reply 3: The assumption is that the target function resides in the Sobolev unit ball $H^s_u([-\pi, \pi]^d)$ under the $L_2$ norm. We consider two scenarios:
> - $d$-demanding case (constant accuracy $\epsilon$ and smoothness $s$): SAQNN exhibits strict asymptotic advantages.
> - $\epsilon$-demanding case (constant $d$ and $s$): SAQNN achieves optimal parameter complexity.
>
> > 4. What are the main new insights of this work compared with existing universal approximation and expressivity results for QNNs?
>
> Reply 4: Compared to existing literature, our work has two key insights:
> - Our model demonstrates a promising pathway for **designing QNNs guided by quantum algorithms (e.g., LCU)**, which usually fully exploit quantum characteristics and may contribute to quantum advantage.
> - Our model achieves **optimal paramter complexity and clear separation of parameters**. In the model raised by Yu et al., trainable parameters and inputs are deeply coupled and it has structural redundancy. In SAQNN, $\boldsymbol{\theta}$ are centralized, $\boldsymbol{x}$ and $\boldsymbol{\phi}$ in each layer are also separated. This modularity may enable advanced optimization techniques such as layer-wise freezing, similar to that in Variational Quantum Eigensolvers (VQE).
>
> > 5. How can the proposed constructive model be translated into practical quantum machine learning algorithms, and how realistic are the claimed advantages under implementation constraints?
>
> Reply 5: We believe that practical ML tasks can be easily adapted to SAQNN. For example, continuous function outputs can be mapped to discrete class intervals and input features (pixels) can be encoded as $\boldsymbol{x}$ for image classification.
>
> On current NISQ devices, the direct deployment of SAQNN indeed faces practical challenges. However, the performance of quantum hardware have been improving at a remarkable pace in recent years, providing a promising outlook that our theoretical advantages could become a reality in the foreseeable future.
>
> > 6. The broader impact would be clearer if the paper better explained how these approximation results inform the design or analysis of practical QML models and algorithms.
>
> Reply 6: As discussed in Reply 4, one can either develop novel architectures based on quantum algorithms or directly deploy SAQNN with suitable spectrum. Besides, SAQNN's modular structure can also help with analysis. For instance, to reduce circuit depth, one can trade expressivity for trainability by replacing state preparation block with a heuristic ansatz. Since this ansatz exclusively controls coefficient magnitudes (see Reply 7), this provides guidance for ansatz selection and simplifies circuit debugging.
>
> > 7. Some additional intuition about the model construction and why it leads to these approximation properties would improve accessibility.
>
> Reply 7: A Fourier series in polar form is $f(\boldsymbol{x}) = \sum |c_{\boldsymbol{j}}| e^{i \text{arg}(c_{\boldsymbol{j}})} e^{i \boldsymbol{j} \cdot \boldsymbol{x}}$. SAQNN deconstructs it and maps it directly onto a quantum circuit:
> - Amplitude $|c_{\boldsymbol{j}}|$: State preparation $P(\boldsymbol{\theta})$ maps coefficient magnitudes 1-1 to the probability amplitudes of the control qubits.
> - Phase $\text{arg}(c_{\boldsymbol{j}})$: Phase injection $S(\boldsymbol{\phi})$ attaches global phases to each magnitude, equipping the model with capacity to learn complex coefficients.
> - Basis $e^{i \boldsymbol{j} \cdot \boldsymbol{x}}$: Spectrum selection $V_r(\boldsymbol{x})$ encodes inputs via rotation gates.
>
> In fact, a standard LCU implements that $U_{\boldsymbol{\theta},\boldsymbol{\phi}}(\boldsymbol{x}) = \begin{pmatrix} \sum_r c_r U_r(\boldsymbol{x}) & * \\\\ * & * \end{pmatrix}$.
>
> We thank the reviewer for helpful comments again. We will make corresponding modifications in revised version of the manuscript.

---

> > ### Author Rebuttal · Reviewer_u459 · 2026-04-05
> >
> > I will update my score.

---

### Official Review · Reviewer_PHzc · 2026-03-12

**Soundness:** 2
**Presentation:** 3
**Significance:** 2
**Originality:** 3
**Overall Recommendation:** 4
**Confidence:** 3

**Summary:**

This paper constructs a quantum neural network based on truncated Fourier series implemented via quantum multiplexors. Authors prove a UAP for multivariate L2 functions. They show advantage over classical FFN for Sobolev function approximation. This advantage does not seem to come from the theorem it prove (though it's properly characterized) but more from the hardware attribute. This does not provide much guidance to FFN practitioner.

**Compliance With Llm Reviewing Policy:**

Affirmed.

**Final Justification:**

NA

**Key Questions For Authors:**

It does not seem to be brand new to discuss the universal approximation property and it is not commonly assumed that Quantum Circuits will be worse than the typical FFN. Is it reasonable to say that Quantum Circuits based model are more general therefore the performance upper limit is definitely better than FFN. It is still non trivial to quantify this result in this paper.

**Limitations:**

This paper is very theoretical and can not be adopted by common machine learning practitioner.

**Strengths And Weaknesses:**

This paper headline is derived from theoretically proposition and very limited experimental validation.

---

> ### Author Rebuttal · Authors · 2026-03-30
>
> We thank the reviewer for these valuable comments on our manuscript. We would like to provide a detailed response to the concerns of the reviewer.
>
> > 1. This advantage does not seem to come from the theorem it prove (though it's properly characterized) but more from the hardware attribute.
> >
> >    It is not commonly assumed that Quantum Circuits will be worse than the typical FFN. Is it reasonable to say that Quantum Circuits based model are more general therefore the performance upper limit is definitely better than FFN. It is still non trivial to quantify this result in this paper.
>
> Reply 1: It is indeed a common intuition that quantum computing processes a distinct advantage because it can compute exponentially many paths simultaneously. However, we clarify that these inherent quantum mechanical properties do not trivially or naturally translate into an algorithmic or modeling advantage over classical counterparts. While renowned quantum algorithms provide strong evidence for quantum supremacy, they are designed for specific problems, and some rely on additional assumptions. Regarding the question of whether Quantum Neural Networks (QNNs) possess stronger expressivity than classical FNNs, while the community is inclined to believe the answer is yes, there has been a lack of fundamental, theoretical results to support this expectation. More importantly, building a constructive quantum model to actually redeem this potential advantage is crucial. This is precisely one of the core contributions of our work. While this advantage is rooted in the hardware attributes (i.e. quantum mechanics), it is also actualized through our well-designed model that fully exploits the power of quantum mechanics, which is often highly challenging. In fact, we have already provided the quantitative results of SAQNN’s circuit cost in $L_2$ approximation of Sobolev function space (see Theorem 2), used to characterize its performance and facilitate the comparison with ReLU-FNNs.
>
> > 2. This does not provide much guidance to FFN practitioner.
>
> Reply 2: The main objective of this work is actually not to provide algorithmic guidance or novel architectures for classical FNN practitioners. Rather, it is a foundational theoretical contribution aimed at the Quantum Machine Learning (QML) community. Our work provides QML researchers with a mathematically reliable blueprint for building QNNs with interpretability. We introduced classical ReLU-FNNs here as a well-established, mathematically mature baseline to quantify the comparative advantages of our SAQNN model.
>
> > 3. This paper headline is derived from theoretically proposition and very limited experimental validation.
>
> Reply 3: We want to emphasize that the main contribution of our work lies in the theoretical approximation capability of quantum neural networks. The numerical simulations presented in our work are intended as proof-of-concept validations to support our theoretical results, rather than as empirical benchmarks on large-scale datasets. In terms of the experiment design, we sincerely refer the reviewer to **REPLY 1** of our rebuttal to **reviewer hY5x** for more detailed reasons.
>
> > 4. It does not seem to be brand new to discuss the universal approximation property.
>
> Reply 4: Though the Universal Approximation Property (UAP) has been explored in prior literature, previous studies have predominantly focused on classical neural networks or quantum-classical hybrid models (as discussed in our Introduction). The expressivity of pure quantum neural networks remain largely under-explored. Furthermore, merely proving the existence of UAP for a quantum model is insufficient, it is crucial to rigorously quantify the required quantum resources. In addition, providing a constructive model is essential. Since real-world quantum hardware typically only supports basic quantum gates (e.g., CNOT and arbitrary single-qubit rotation gates), an explicit construction directly dictates whether a model is practically implementable on quantum devices. As we highlighted in Reply 1, these make up the core contributions of our work.
>
> > 5. This paper is very theoretical and can not be adopted by common machine learning practitioner.
>
> Reply 5: We acknowledge that our current work focuses mainly on establishing strict theoretical foundations, but we argue that this is a vital prerequisite for reliable practical adoption of QML. Historically, foundational classical machine learning milestones such as the original proofs of the UAP were also highly theoretical before becoming the bedrock of modern, practitioner-friendly models. In our constructive model, we are essentially providing the exact specifications needed by tool developers. As the quantum computing industry matures and moves toward standardized software stacks, such explicitly designed architectures can be encapsulated into high-level libraries.

---

> > ### Author Rebuttal · Reviewer_PHzc · 2026-04-05
> >
> > This rebuttal has resolved all my concerns.

---

### Decision · Program_Chairs · 2026-04-30

**Decision:**

Accept (regular)

**Comment:**

This paper is a theoretical one that studies universal function approximation by quantum neural networks. In particular, it proved that quantum neural networks have asymptotic advantages over the best classical feed-forward neural networks in terms of circuit size and achieves optimal parameter complexity when approximating Sobolev functions under $\ell_{2}$ norm.

The initial reports by reviewers pointed out the theoretical guarantees are solid, but there were some concerns about the practicality, comparison to existing works, etc. The authors made decent rebuttals for clarifying these points, and all scores converge to accepts unanimously. The decision is hence acceptance - the ICML 2026 audiences will benefit from such a theory result and understanding the difference between quantum and classical neural networks.